# Evaluation of Microglia/Macrophage Cells from Rat Striatum and Prefrontal Cortex Reveals Differential Expression of Inflammatory-Related mRNA after Methamphetamine

**DOI:** 10.3390/brainsci9120340

**Published:** 2019-11-25

**Authors:** Joanne S. Kays, Bryan K. Yamamoto

**Affiliations:** Department of Pharmacology and Toxicology, Indiana University School of Medicine, Indianapolis, IN 46202, USA; kaysj@iupui.edu

**Keywords:** RNA sequencing, microglia, macrophage, methamphetamine, inflammation

## Abstract

RNA sequencing (RNAseq) can be a powerful tool in the identification of transcriptional changes after drug treatment. RNAseq was utilized to determine expression changes in Fluorescence-activated cell sorted (FACS) CD11b/c+ cells from the striatum (STR) and prefrontal cortex (PFC) of male Sprague-Dawley rats after a methamphetamine (METH) binge dosing regimen. Resident microglia and infiltrating macrophages were collected 2 h or 3 days after drug administration. Gene expression changes indicated there was an increase toward an overall pro-inflammatory state, or M1 polarization, along with what appears to be a subset of cells that differentiated toward the anti-inflammatory M2 polarization. In general, there were significantly more mRNA expression changes in the STR than the PFC and more at 2 h post-binge METH than at 3 days post-binge METH. Additionally, Ingenuity^®^ Pathway Analysis along with details of RNA expression changes revealed cyclo-oxygenase 2 (COX2)-driven prostaglandin (PG) E2 synthesis, glutamine uptake, and the Nuclear factor erythroid2-related factor 2 (NRF2) canonical pathway in microglia were associated with the binge administration regimen of METH.

## 1. Introduction

Methamphetamine (METH) is known to induce neuroinflammation. Resident glial cells, including microglia [1,2,3] and astrocytes [4,5,6] in the brain are implicated in contributing to this METH-induced neuroinflammation via production and release of a variety of inflammatory mediators, such as interleukin (IL) 1β [7,8], IL6 [9,10], tumor necrosis factor (TNF) α [11], C-C motif chemokine ligand 2 (CCL2) [12], reactive oxygen species (ROS) [13,14], and nitric oxide [15]. Inducible enzymes that can lead to the production of additional inflammatory mediators in these glial cells, such as COX2 [16,17], have also been implicated. However, the specific cellular origin of these pro-inflammatory mediators has yet to be defined.

The canonical role of central nervous system (CNS)-resident microglia is to serve as sentinels to detect disturbances in the chemical environment around neurons and to respond to these disturbances. In their response to locally detected lipopolysaccharide (LPS) or interferon-gamma (INF-γ), microglia can take on the role known as the M1 phenotype. As an activated M1 microglia, the microglia produce and release pro-inflammatory mediators. In contrast, microglia can take on an M2 phenotype if stimulated locally by IL4 or IL13. In this state of activation, the microglial cell produces and releases mediators such as IL10, which can serve as an anti-inflammatory signal to initiate recovery from an insult.

The current study focused on the role of microglia at 2 h and 3 days after administration of neurotoxic binge doses of METH to evaluate time-dependent, acute, and persistent changes in gene transcription within microglia and infiltrating macrophages. We evaluated changes in gene transcription in the rat striatum (STR) and prefrontal cortex (PFC), areas associated with controlling voluntary movement, reward, motivation, and executive functions such as planning and decision-making [18,19,20,21], all of which are affected by METH [22,23,24,25].

## 2. Materials and Methods

### 2.1. Animals

All experiments were conducted with approval of the Institutional Animal Care and Use Committee at the Indiana University School of Medicine and performed in American Association for Laboratory Animal Care (AALAC)-approved facilities. Male Sprague-Dawley rats were purchased from Envigo (Indianapolis, IN, USA) and received food and water ad libitum. After a period of acclimation, the rats were randomly divided into either control (saline) or treated (METH) groups. Methamphetamine (METH) HCl was from Sigma-Aldrich (St. Louis, MO, USA, Catalog #M-8750). A 10 mg/mL METH solution was prepared the day of injections in sterile-filtered 0.9% *w/v* sodium chloride (saline). Rats received intraperitoneal (i.p.) injections of either saline or 10 mg/mL METH at a volume equal to 1 mL/kg (10 mg METH/kg body mass). Injections were given at 2 h intervals over 8 h for a total of 4 injections. Rats were housed individually during the injections in plastic shoebox-style containers with a wire lid. Food and water were available ad libitum. A plastic grate was placed on top of the bedding inside the container to prevent aspiration of bedding material. Body temperature was monitored at 15–30 min intervals throughout the injection period and for 1–2 h after the final injection via a subcutaneously-implanted temperature transponder (Bio Medic Data Systems, Inc., Seaford, DE, USA, Catalog # IPTT 300). If a rat’s body temperature exceeded 40 °C, measures were taken (oscillating fan, ice packs on cage lid) to prevent hyperthermia. At either 2 h or 3 days after the last i.p. injection, each rat was deeply anesthetized with an i.p. injection of a ketamine (Henry Schein, Indianapolis, IN USA, Ketathesia^TM^ 100 mg/mL)/xylazine (Akorn Pharmaceuticals, Decatur, IL, USA, AnaSed^®^ Injection 20 mg/mL) 70%/30% *v/v* mix. Once a rat had no response to a toe-pinch, the chest cavity was opened and 0.35 mL heparin (Sagent Pharmaceuticals, Inc, Schaumburg, IL, USA, 1000 USP units/mL) was injected into the left ventricle, followed by perfusion with ice-cold phosphate-buffered saline (PBS). Blood and PBS exited the body through an incision made in the right atrium. Perfusion continued for 5 min at a flow rate of 20 mL/min. We chose to perfuse with PBS prior to brain extraction to reduce the contribution of circulating macrophages contained within the brain’s capillaries, thereby restricting our collection to Cd11b/c+ cells that reside within the brain parenchyma.

The rat was decapitated via guillotine and the whole brain was removed from the skull. The brain was bisected on ice into its right and left hemispheres via RNase Away^®^-treated (Thermo Fisher, Waltham, MA, USA, #7005-11) razor blade and the striatum and prefrontal cortex were blunt-dissected from each hemisphere. The two pieces of the same region were combined into a single DNase-free RNase-free 1.5 mL tube containing 200 μL of ice cold dissociation buffer (see below).

The number of rats per group were: 2 h post saline STR (6), 2 h post METH STR (8), 2 h post saline PFC (4), 2 h post METH (5). The respective numbers for the 3 day groups were 6, 7, 6, and 7.

### 2.2. Buffers and Antibodies for Single-Cell Suspensions

For the mechanical dissociation of STR and PFC to a single-cell suspension, several buffers were prepared. Diethyl pyrocarbonate (DEPC)-phosphate-buffered saline (PBS) pH 7.4 was prepared using DEPC-treated water (Thermo Fisher, AM9922) containing 150 mM sodium chloride (Santa Cruz Biotechnology, Dallas, TX, sc-203274B), 1.5 mM sodium phosphate monobasic (Sigma-Aldrich S-9638), and 17.5 mM sodium phosphate dibasic (Thermo Fisher, S374-1). Dissociation buffer was prepared with DEPC-PBS supplemented with RNase-free DNase (Qiagen, Hilden, Germany) 79254, stock = 2.72 units/μL, final concentration in DEPC-PBS = 40 units/mL) and RNasin^®^ Plus RNase Inhibitor (Promega, Madison, WI, USA, N2615, stock = 40 units/μL, final concentration in DEPC-PBS = 80 units/mL). Blocking buffer consisted of DEPC-PBS with 1% *w/v* bovine serum albumin (BSA) (Sigma-Aldrich, A3059), 80 units/mL RNasin^®^ Plus RNase Inhibitor and 5% *v/v* normal mouse serum (Sigma-Aldrich, M5905). Blocked cells were divided, with a fraction of the cells incubated with phycoerythrin (PE)-labeled isotype IgG2a control antibody (Abcam, Cambridge, MA, USA, ab91363) and the remainder of the cells incubated with mouse monoclonal phycoerythrin (PE)-labeled antibody to CD11b/c (Abcam, ab112239) in blocking buffer. For fluorescence activated cell sorting (FACS), stained and washed cells were resuspended in DEPC-PBS with 1% *w/v* BSA and 80 units/mL RNasin^®^ Plus RNase Inhibitor (Promega).

### 2.3. Mechanical Dissociation

Keeping the tissue-containing tube on ice, an RNase Away^®^-treated plastic pestle designed to fit a 1.5 mL tube was used to gently disrupt the tissue to smaller pieces. Another 300 μL ice cold dissociation buffer was added to a final volume of 500 μL. Any remaining tissue chunks were mashed. The suspension was filtered through 100 micron mesh into a clean 1.5 mL DNase-free, RNase-free tube on ice. The filter was washed with another 500 μL dissociation buffer into the same tube. The combined filtrate was centrifuged 5 min 2500× *g* at 4 °C to pellet the cells. After aspirating the supernatant, the cells were blocked and incubated with PE-labeled CD11b/c+ antibody.

### 2.4. Blocking and Staining Rat Striatal Cells

One mL of blocking buffer was added to each cell pellet and the cells were resuspended by gentle pipetting. The tubes were closed and rocked for 10 min at room temperature (RT). Fifty μL was removed from each sample and combined in a common tube for gating of cells via flow cytometry (FACS). Another 50 μL was removed from each sample to combine into a negative control for PE nonspecific binding with the PE-labeled isotype control antibody. The remaining 900 μL of cells from each sample were used for PE-labeled CD11b/c antibody labeling. All tubes were spun for 5 min at 2500× g 4 °C to pellet the cells. Supernatant was discarded. Cell pellets from each sample to be incubated with PE-labeled CD11b/c antibody were resuspended in 500 μL of blocking buffer containing 5 μL (0.5 μg) of antibody. A proportionate amount of PE-labeled isotype control antibody was incubated with the resuspended cell pellet for nonspecific PE binding. The resuspended cell pellet that would not receive any antibody was incubated in the same volume of blocking buffer as the isotype control pellet. All tubes were wrapped in foil to protect the fluorescent label from light and rotated end-over-end overnight at 4 °C.

### 2.5. FACS and RNA Isolation

The next morning, the cells were collected by centrifuging the tubes for 5 min at 2500× *g* 4 °C. After aspirating the supernatant, each cell pellet was washed twice by resuspending it in 500 μL DEPC-PBS, centrifuging, and aspirating the DEPC-PBS. The final pellet was resuspended in DEPC-PBS + 1% BSA + RNasin^®^ Plus RNase Inhibitor for cell sorting on BD Biosciences FACSAria flow cytometers. Figure 1 is representative of the gating strategy used to isolate PE-CD11b/c+ cells (P3). The collected CD11b/c+ cells were pelleted by centrifugation at 16,000× *g* for 5 min at 4 °C. After aspirating the supernatant, the pellet was resuspended in 50 μL Solution XB (Arcturus^TM^ PicoPure^TM^ RNA Isolation Kit, Thermo Fisher, KIT0204) and incubated at 42 °C for 30 min. To remove any insoluble material, the tubes were spun at 3000× g for 2 min and the cleared supernatant was transferred to a clean DNase-free RNase-free 1.5 mL tube. Total RNA was isolated as per the instructions provided with the Arcturus^TM^ PicoPure^TM^ RNA Isolation Kit instructions (ThermoFischer), including optional on-column DNase treatment.

### 2.6. Clontech SMART-Seq V4 Ultra Low Input RNA Methods for Illumina HiSeq 4000 Sequencing

Purified total RNA was first evaluated for its quantity and quality, using Agilent Bioanalyzer 2100 (Agilent, Santa Clara, CA, USA). One nanogram of total RNA per sample was used for library preparation. cDNA was first synthesized using SMART-Seq v4 Ultra Low Input RNA Kit for Sequencing (Takara Clontech Laboratories, Inc., Mountain View, CA, USA). Dual indexed cDNA library was then prepared using Nextera XT DNA Library Prep Kit (Illumina, Inc., San Diego, CA, USA). Each library was quantified and its quality accessed by Qubit and Agilent Bioanalyzer, and multiple libraries were pooled in equal molarity. The average size of the library insert was about 150–200 b. Five microliters of 2 nM pooled libraries per lane were then denatured, neutralized, and applied to the cBot for flow cell deposition and cluster amplification, before loading on to HiSeq 4000 for 75 b paired-end sequencing (Illumina, Inc.). Each lane generated approximately 300–350 million reads. A Phred quality score (Q score) was used to measure the quality of sequencing. More than 90% of the sequencing reads reached Q30 (99.9% base call accuracy).

### 2.7. Sequence Alignment and Gene Counts

The sequencing data were first assessed using FastQC (Babraham Bioinformatics, Cambridge, UK) for quality control. All sequenced libraries were mapped to the rat genome (mm10) using STAR RNA-seq aligner [26] with the following parameter: “--outSAMmapqUnique 60”. The reads distribution across the genome was assessed using bamutils (from ngsutils) [27]. Uniquely mapped sequencing reads were assigned to mm10 refGene genes using featureCounts (from subread) [28] with the following parameters: “-s 2 –p –Q 10”. Quality control of sequencing and mapping results were summarized using MultiQC [29]. The data were normalized using the TMM (trimmed mean of M values) method. Differential expression analysis was performed using edgeR [30,31]. False discovery rate (FDR) was computed from *p*-values using the Benjamini–Hochberg procedure.

### 2.8. Ingenuity^®^ Pathway Analysis

Ingenuity^®^ Pathway Analysis (IPA^®^) software (Qiagen, Germantown, MD, USA) was employed to analyze relationships between genes that exhibited significant differential expression (FDR < 0.05) as a result of binge METH treatment.

## 3. Results

### 3.1. General Information from RNAseq

As seen in Table 1, we obtained identifiable sequences for around 10,000 genes in each brain region and at each time point. STR appears to yield more genes that were significantly changed by METH treatment than the PFC at both time points. The 2 h post-METH time point revealed the STR had 4483 genes and the PFC had 3118 genes that were significantly changed by METH. By the three-day post-METH time point, only 714 genes reached a significant level of difference in expression and all of those genes were in the STR. Of the 7601 genes significantly changed by METH at the 2 h time point in the combined STR and PFC, 2483 genes were common to both regions, with 1255 of those genes significantly upregulated in both STR and PFC and 1223 significantly downregulated in both regions, considering all significant fold-changes (Figure 2a). The remaining five common genes are listed in Table 2. Four genes (*Plau*, *Itgb3*, *Tmem120a*, *Msto1*) were significantly downregulated in PFC while significantly upregulated in the STR and 1 gene (*Grik5*) was significantly upregulated in the PFC while significantly downregulated in the STR. When fold-change values are restricted to those at least two-fold (Figure 2b), the number of significantly changed genes common to both the STR and PFC is reduced from 2483 to 779. Of those 779 genes, 435 are upregulated in both STR and PFC, while 342 are downregulated. *Plau* and *Grik2* remain present as genes significantly expressed in opposing directions between the two brain regions. Note that the majority of the significant changes were less than two-fold up or down across all comparisons.

There were 355 genes in the STR that were significantly changed at both the 2 h and 3 day time points. Of these 355 genes, 96 were upregulated at both time points, 70 were downregulated, 80 were up- then down-regulated and 109 were down- then upregulated. When restricted to fold changes >2 or <−2, the number of genes fell to 6, 6, 5, and 4, respectively (Table 3).

Table 4 lists the IPA^®^ results of all genes with an FDR <0.05 in the STR or PFC at each time point regardless of the degree of fold-change. At the 2 h time point, changes in gene expression induced by binge METH in the STR and in the PFC share unfolded protein response as one of the top five canonical pathways. Lipopolysaccharide and platelet-derived growth factor beta-beta (PDGF BB) are common to both regions as the top five upstream regulators. STR and PFC share three of the five top molecular and cellular functions (cell death and survival, cellular movement, cellular development). However, at 3 days post-METH, the IPA^®^ of the gene expression patterns dramatically changed in the STR. At 2 h post-METH, the apparent emphasis of cellular functions was focused on more “macro” processes such as cell development, movement, and death/survival. By 3 days post-METH, the cellular function switched to more “micro” processes devoted to transcription and translation. Since there were no longer any detectable gene expression changes in the PFC at the 3 day time point, no IPA^®^ at this time point could be performed.

Table 5 summarizes IPA^®^ results using the RNAseq data for the 355 genes significantly expressed in STR at both 2 h and 3 days post-binge METH with no restrictions on the fold-change. For the subset of genes in the STR significantly upregulated at both time points, there is an emphasis on pathways connected to gene and protein expression that influence cell morphology and cell-to-cell interaction after binge METH. Pathways that reveal a consistent downregulation over time in the STR after binge METH involve tryptophan degradation, nicotinamide adenine dinucleotide (NAD) biosynthesis, and leukotriene biosynthesis. It appears that functions relating to DNA and cell replication are slowed over time by this subset of genes. For the 80 genes initially upregulated significantly at 2 h and then downregulated significantly at 3 days, regulation by NFkB may be involved as three of the five top upstream regulators are IKBKE, RELB, and NFKB2. For the 109 genes initially significantly downregulated at 2 h then upregulated at 3 days, NAD salvage was the top canonical pathway identified.

### 3.2. RNA Sequencing Data for the Cited Top 25 Genes Enriched in Microglia

Butovsky et al. [32] did an expansive study to create an RNA profile of microglia compared to other monocyte lineage cells and cell lines as well as oligodendrocytes, astrocytes, and neurons. Table 6 presents the current study’s data for the Butovsky et al. list of top 25 genes enriched in naïve CD11b+ FRCLS+ adult mouse whole brain cells (microglia). Since these 25 genes represent a profile of expression that may serve to define microglia vs. other cell types, we did not necessarily expect to find that METH treatment would change the expression levels of these genes. However, 12 of these 25 genes were significantly changed by METH at the 2 h time point with four genes upregulated and eight genes downregulated. Note that *P2ry12* was significantly downregulated as reported in association with microglial activation [33]. While we did not study any other cell type other than CD11b/c+ for this study, these “microglia enriched” genes are very highly expressed compared to the average expression level of all 10,839 detected genes. The average log2cpm (counts per million) for all genes was around 3.9 while the average log2cpm for the 24 detected genes listed in Table 6 was close to 9.

### 3.3. Comparison of RNA Sequencing Data to Gene Markers of Microglia vs. Macrophages

Table 7 contains the average log2cpm expression from our 2 h post-saline or post-METH STR or PFC rat microglia of the top 25 genes designated by Hickman et al. [34] as being genes unique to either microglia isolated via FACS from mouse whole brain or to peritoneal macrophages collected by lavage. While 6 of the cited top 25 unique mouse microglial genes were not found in our set of data, the remaining 19 were present with a log2cpm average of 7.5 to 8.5. Ten of the 25 unique mouse macrophage genes were not present in our data on rats. The remaining 15 genes had an overall average log2cpm around 2.0, a much lower expression than that of the microglial genes.

### 3.4. Comparison of METH-Induced RNAseq Gene Expression Changes to Reported M1 vs. M2 Polarization Markers

Microglia can be activated from a resting state to an activated state and subsequent polarization by a variety of mediators released by a variety of insults. In broad terms, M1-polarization is characterized as activated microglia which produce and release inflammatory mediators themselves, whereas, M2-polarization is associated with activated microglia producing and releasing mediators which act in a neuroprotective role. Table 8 compares RNAseq data from our 2 h post-METH STR and PFC to 47 genes cited in literature reviews as indicating M1 or M2 polarization of microglia [35,36,37,38,39]. Of these 47 genes, we found 12 genes were significantly changed by binge METH in the same direction as reported in the literature (shaded in green). Five of these 10 genes (*Cd32*, *Cd80*, *CXCL1*, *IL1a*, *IL1b*) were associated with M1 polarization while six of the 10 genes (*Cd204*, *Dectin1*, *Gal3*, *Ifg1*, *Il1ra*, *Mrc1*) were associated with M2 polarization. *Ptgs2* (COX2) was also upregulated and associated with M1 and M2 polarization [35].

METH is known to upregulate COX2 expression at the RNA [40] and protein levels [17] in rodent brains. One consequence of COX2 upregulation is the enhanced production of PG’s [41,42]. Therefore, we examined the effects of binge-METH on the RNA expression levels of key molecules involved in PG synthesis. As shown in Figure 3, genes whose proteins are involved in the release of arachidonic acid (AA) from cell membranes and the subsequent synthesis of PG’s are upregulated 2 h after binge-METH treatment. Additionally, there appears to be a diversion of AA to PG’s over the cysteinyl leukotrienes (LT) via the corresponding downregulation or lack of detection of genes for the enzymes responsible for LTC4, D4, and E4 synthesis. LTA4 could be directed instead to the production of LTB4 via upregulation of *Lta4h*.

Figure 4 illustrates that the genes that could yield proteins which enable the specific synthesis of inflammatory mediator PGE2 from PGH2, along with two of the four receptors for PGE2, are upregulated. This hints that exposure to binge METH could elicit an autocrine mechanism to positively feedback onto an inflammatory pathway within CD11b/c+ cells. Since METH treatment does not change *Alox5* or *Alox12* expression, generation of 5*S*- and 12*S*-HETE (Hydroxyeicosatetraenoic acid) may not change, but their ability to possibly function in an autocrine fashion would likely be reduced since the genes for the corresponding receptors, *Oxer1* and *Gpr31*, are respectively not detected or significantly downregulated. *Alox15* is not detected; however, expression of the gene for a 15*S*-HETE receptor, *Ppard*, is upregulated.

## 4. Discussion

Significant changes in gene expression were found in microglial cells of rat STR and PFC following a binge METH dosing paradigm. These changes are detectable at 2 h after the last dose of METH, with some changes persisting for up to 3 days. About one third of genes changed by binge METH at 2 h after METH are shared by the STR and PFC. However, the microglia from the STR appear to exhibit longer-lasting changes than microglia from the PFC given that the PFC had no significantly changed genes at 3 days after METH. Since genes attributed as being unique to macrophages are either not detected or are present only at very low levels, peripheral macrophages appear to have minimal contribution to the number of CD11b/c+ cells collected from rat STR or PFC at 2 h post saline or METH.

Activation of microglia from the steady-state involves differential gene expression patterns that direct the cell into one of at least two polarizations, the most common designations being M1 and M2. At 2 h post-binge METH, microglia appear to be transitioning from a resting state to an M1 inflammatory-reactive state. *Il1b* is upregulated 2 h after the last dose of binge METH in both the STR and PFC and emerges as a key marker of such activation. Moreover, *P2ry12* was downregulated in both brain regions at 2 h post-binge METH and is consistent with the finding that microglial activation is associated with its downregulation [33]. These findings are in contrast to other known markers of M1 activation (*Il6*, *Tnf*, *Ccl2*, *iNos*, *Nox*) that were not changed after METH and suggest that microglia are only in the early stages of responding to the insult of binge METH. Thus, this study provides insight as to how soon microglia respond to binge METH and that *IL1β* is an early inflammatory mediator produced by microglia after METH exposure. These findings add to a growing body of literature that microglial activation can be further differentiated beyond M1 and M2 that depend on the type of insult [44,45]. More specifically, microglial activation states also differ in rodent models of aging [34], retinal degeneration [38], ischemia [37,46], Alzheimer’s disease [47], Parkinson’s disease [48,49], and binge alcohol consumption [39].

While *Tnf* RNA expression was not changed in response to binge METH treatment, there were increases in the expression of a specific TNF superfamily member *Tnfsf*9. *Tnfsf*9 was upregulated 19.5-fold and 11.5-fold in the STR and PFC, respectively, at 2 h post-binge METH. TNFSF9 is also known as CD137L or 4-1BBL, a transmembrane cytokine expressed on antigen presenting cells (APC), including macrophages [50]. TNFSF9 interacts with its receptor, TNFRSF9 on activated T-cells, and can generate a signaling cascade. Yeo et al. [51] demonstrated that interaction of TNFSF9 on BV2 and N9 cells as well as on primary microglia resulted in responses typically associated with activation of microglia to an inflammatory condition, including generation of reactive oxygen species (ROS). Thus, binge METH may produce a specific upregulation in this one TNF superfamily member which contributes to an M1 inflammatory activation state.

IL1β production after METH can be induced by LPS [52,53] that in turn, can increase the release of PGE2 in a manner attenuated by COX inhibitor indomethacin [54], suggesting of a role for prostaglandins and the arachidonic acid cascade. Furthermore, PGE2 acting through the EP2 receptor was essential for pro-IL1β production induced by LPS in mouse bone-marrow derived macrophages [55]. Along these lines, Figure 3 and Figure 4 illustrate that METH changes expression for a number of genes involved in directing arachidonic acid metabolism toward PGE2 production that could be dependent on LPS derived from a leaky gut. In fact, Persons et al. [56] showed that METH self-administration in Fischer 344 rats resulted in lower expression of gut tight junction proteins claudin-1 and ZO-1 as well as a morphological disorganization of these proteins in the colon. Additionally, human METH users suffer from bowel ischemia which is associated with loss of gut integrity [57,58] that may lead to the entry of LPS into the brain parenchyma as shown in rats following a METH-induced compromise of the blood-brain-barrier [59]. Thus, high doses of METH may promote PGE2 production in brain microglia that exacerbates LPS-induced IL1β production. The enhanced expression of *Ptger2* suggested by our current RNAseq data could further promote this cycle by allowing PGE2 to act in an autocrine fashion to induce COX2 expression via EP2 receptor activation [60]. This feed-forward cycle involving LPS, EP2, and COX2 is supported by the observation that EP2-deficient mice have reduced serum levels of inflammatory mediators after the LPS challenge [61]. Moreover, activation of EP2 in cultures of newborn Sprague-Dawley rat cortical microglia with PGE2 or butaprost upregulates mRNA’s encoding COX2 and IL1β, as well as iNOS, and IL6 [62].

*Il1r2* had a sharp increase in expression 2 h post-binge METH in both the STR (39-fold) and PFC (131-fold). IL1R2 protein is a membrane-bound decoy receptor for IL1 expressed in mouse neutrophils which under LPS-stimulation was increased to its cleaved soluble form by ADAM17 [63]. Johannson et al. [64] found that *Il1r2* RNA expression was enhanced in microglia isolated from adult mouse brains exposed to Aβ peptide. The large increase in *Il1r2* expression in microglia 2 h post-binge METH indicates that microglia are similarly responsive to METH. Interestingly, *Il1r2* expression was significantly decreased in microglia from mouse brains exposed to Aβ peptide in EP2 knock-out mice vs. wild type EP2 mice [64]. In addition, EP2-/- mice had enhanced microglial Aβ phagocytosis characteristic of M2 anti-inflammatory microglia and a lack of Aβ-activated paracrine neurotoxicity [65]. It is possible that enhanced *Ptger2* expression induced by binge METH treatment may serve to enhance *Il1r2* expression and suppress transition toward an M2 anti-inflammatory state in the early hours after METH.

The production of the cysteinyl LT’s from AA may be diminished as illustrated by the gene expression results in Figure 3. However, it is noted that *Alox5* is present and *Alox5ap* is upregulated which could still result in LTA4 synthesis. LTA4 is the precursor to not only the cysteinyl LT’s but also to LTB4. The gene for the enzyme that converts LTA4 to LTB4, *Lta4h*, is significantly upregulated. LTB4 longevity could be enhanced via the downregulation of *Ptgr1*, the gene for prostaglandin reductase 1, a key enzyme for inactivating LTB4 as an inflammatory mediator [66]. In models of LPS-induced septic shock, Collin et al. [67] and Kwon et al. [68] demonstrated that inhibition of ALOX5 and antagonism LTB4 receptors, respectively, reduced multiple LPS-induced effects. Saiwai et al. [69] used a model of spinal cord injury and reported that LTB4 receptor inhibition suppressed leukocyte infiltration and attenuated the inflammatory reaction. Since no RNA for LTB4 receptors was detected in the RNAseq data, the potential LTB4 production in microglia from METH-treated rats would not act in an autocrine fashion, but most likely be released to stimulate LTB4 receptors on other cells. Thus, it appears that LTB4 could be a potential contributor to METH-induced inflammation.

The IPA^®^ results showed that there is an enhancement of the NRF2 (Nuclear factor erythroid 2-related factor 2) (NFE2L2)-mediated canonical pathway in the STR microglia at 2 h post METH. METH is known to induce production of reactive oxygen species [13,70]. The NRF2-mediated pathway is a cellular defense mechanism against oxidative stress (Figure 5). It is noteworthy that among the elements which can activate this pathway are drugs, cytokines, prostaglandins, and bacterial infections. NRF2 appears to be a key connection in the upregulation of *Hsp*’s *22 (Hspb8)*, *40* and *90*, *Stip1*, *Cct7*, *Nqo*, *Gclc*, *Hmox-1*, *Ftl*, *Fth1*, *Sod1*, *Txnrd1, small Maf’s* (*Maff*, *Mafg*, *Mafk*) as well as *Nrf2* itself. Small MAF’s and NRF2 dimerize to form transcriptional activators, suggesting another point of positive feedback in STR after METH and further illustrates the different aspects of microglia activation. More specifically, *Hsp-22*, *Hsp-40* family member genes, and *Hsp90* are significantly upregulated 2 h after METH (up to 37-fold) as are *Stip1* (16-fold) and *Fkbp5* (2-fold). HSP’s are critical to protein folding and maturation. The protein product STIP1 is a co-chaperone for HSP90 and serves as a scaffold for the interaction of HSP90 with HSP70. HSP90 ATPase activity is inhibited by PTGES3 by its binding to the closed conformation of dimeric HSP90 and delaying release of a client protein [71]. *Ptges3* is significantly upregulated over 2-fold in both the STR and PFC 2 h post METH (Appendix A). Thus, protein products of *Ptges3* may serve two roles by promoting the synthesis of PGE2 and by influencing the maturation and folding of proteins.

CCT‘s, chaperonin containing TCP1 complexes, are also involved in the folding of proteins, including actin and tubulin. *Cct7*, the gene for molecular chaperone subunit 7 of CCT, was significantly upregulated in both the STR and PFC, 3.0- and 2.8-fold, respectively. LPS has been shown to stimulate rearrangement of actin within primary cultures of peritoneal rat macrophages [72] and rat neonatal microglia [73], which can be associated with changes in microglial morphology associated with activation. Thus, upregulation of *Cct7* may be associated with the folding of cytoskeletal proteins linked to changes in morphology of microglia after activation.

Several genes upregulated by NFR2 are associated with detoxification of reactive species. *Nqo* upregulation would result in expression of a protein that serves to detoxify reactive quinone species. Dopamine quinone formation is induced by METH in dopaminergic neurons [74,75] and can activate microglia [76]. *Gclc* upregulation can lead to enhanced expression of glutamyl cysteinyl ligase which is the rate limiting enzyme in the synthesis of glutathione, a key molecule involved with the detoxification of reactive species. Upregulation of *Gclc* and *Sod1* could be associated with the microglial uptake of glutamine. As shown in Figure 6, several genes for solute carriers involved with glutamine uptake are significantly upregulated in microglia. It is known that METH causes the release of glutamate from neurons [77,78] that is ordinarily taken up by astrocytes [79] and converted to glutamine [80], released into the synaptic space and subsequently taken up by neurons where it is converted back glutamate. The RNAseq data indicate that at 2 h post METH, the microglia may be primed to enhance their uptake of glutamine and convert it ultimately to reduced glutathione via the 9-fold increase in *Gclc* (Figure 6). Such a conversion along with the increase in *Sod1* may be associated with a response by microglia to detoxify reactive species but appear to be countered by the downregulation of genes for the enzymes such as glutathione-S-transferases (GST), glutathione peroxidase 8 (GPX8), and catalase (CAT) needed to complete the detoxification cycle. Consequently, the balance of effect could lead to the release and accumulation of pro-oxidant species such as H_2_O_2_ [81,82] to produce the neurotoxicity to METH.

Several reviews point to the evolution from thinking of microglia as either simply M1 or M2 phenotypes to a spectrum of phenotypes or even sub-populations of cells specialized in their response to a given stimulus [44,84,85]. One might envision a subset of microglia poised to respond to the insult of toxic METH by locally generating COX2 expression and subsequently PGE2 production and release. These cells could amplify their initial PGE2 response via autocrine signaling through enhanced EP2 expression, while also permitting activation of a separate subset of microglia poised to respond to PGE2 via upregulated expression of EP4 receptors to enhance phagocytosis [86], suppress expression of inflammatory mediators [87,88] and enhance expression of enzymes to detoxify reactive oxygen species via NRF2 [89]. METH may serve to disrupt this coordinated effort by the incomplete detoxification of reactive species when *Gclc* and *Sod1* are upregulated but *Gpx8* and *Cat* are downregulated. Single-cell RNAseq could be a useful tool to determine if such subpopulations of cells could exist.

There are only six genes upregulated >2-fold at both 2 h and 3 days in the STR after METH. The gene for Betaine/GABA transporter 1, *Slc6a12*, is one of these genes (Table 3, Appendix A). In a review by Kempson et al. 2014 [90], the role of this particular GABA transporter in the brain is described as controversial since its expression levels relative to other GABA transporters are very low. While expression levels of genes for GABA transporters in our RNAseq data from microglia are modest, *Scl6a12* is expressed at similar basal levels to the only other detected gene for a GABA transporter, *Slc6a1*, which is not changed in the STR at either time point by METH. The role of potential enhanced uptake of an inhibitory neurotransmitter by microglia for at least 3 days after METH has yet to be explored.

Another gene that remains upregulated at 3 days after METH in the STR is *Csf1* or macrophage colony stimulating factor (Table 3, Appendix A). CSF1 and activation of its receptor CSF1R, regulates the survival, proliferation, and differentiation of macrophages and microglia during development [91]. The gene for CSF1R is highly expressed in all CD11b/c+ cells analyzed but its expression is not changed by METH (Appendix A). The alternative ligand for CSF1R, IL34, is significantly upregulated 3-fold at 3 days post METH in the STR (Appendix A) and also can stimulate CSF1R. Thus, both ligands elicit a pro-inflammatory phenotype in microglia [92] that persists for at least 3 days after METH.

## 5. Conclusions

The present RNA sequencing results indicate that CD11b/c+ cells isolated from rat STR and PFC at 2 h post-binge METH appear to exhibit an RNA profile which indicates an overall inflammatory activation state. It is noted that genes for some classical M1 markers were not expressed at higher levels or even detected after exposure to METH. This may be due to the 2 h early time point chosen and/or to the nature of how METH activates microglia. These cells appear to recover from the METH-induced inflammatory profile rather quickly. By 3 days post-METH, there are no METH-induced transcriptional changes in the PFC and the changes greater than 2-fold in the STR have dropped to only 78 genes. It remains to be determined if the changes in RNA may be initiators of more prolonged and sustained response by inflammatory proteins. A key pathway that appears to be involved in the inflammatory nature of these cells in response to METH is the arachidonic acid metabolism cascade which favors the production of PGE2. Consequently, PGE2 could be released from the microglia to activate EP receptors on adjacent cells or to act potentially in an autocrine fashion as suggested by the METH-induced upregulation of EP2 and EP4 receptors. Thus, the COX2-PGE2-EP2 cycle may enhance LPS-induced IL1β production and a COX2-PGE2-EP4-NRF2-mediated cycle may drive the response to the oxidative stress produced by METH.

## Figures and Tables

**Figure 1 brainsci-09-00340-f001:**
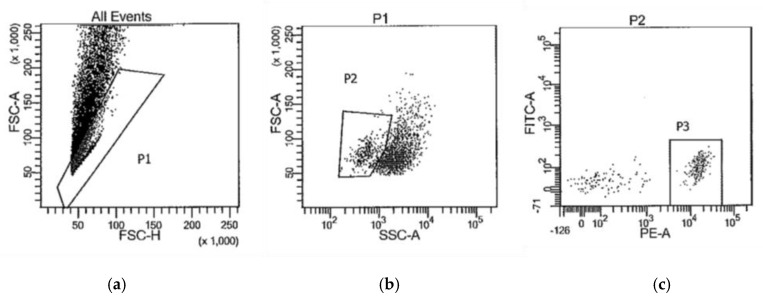
Representative gating strategy to collect CD11b/c+ cells (P3) by FACS for RNA sequencing. (**a**) forward scatter height vs. forward scatter area plot to isolate single cells (P1) from doublets; (**b**) side scatter vs. forward scatter plot to isolate cells (P2) from debris; (**c**) fluorescence intensity plot to isolate Cd11b/c-PE labeled cells (P3).

**Figure 2 brainsci-09-00340-f002:**
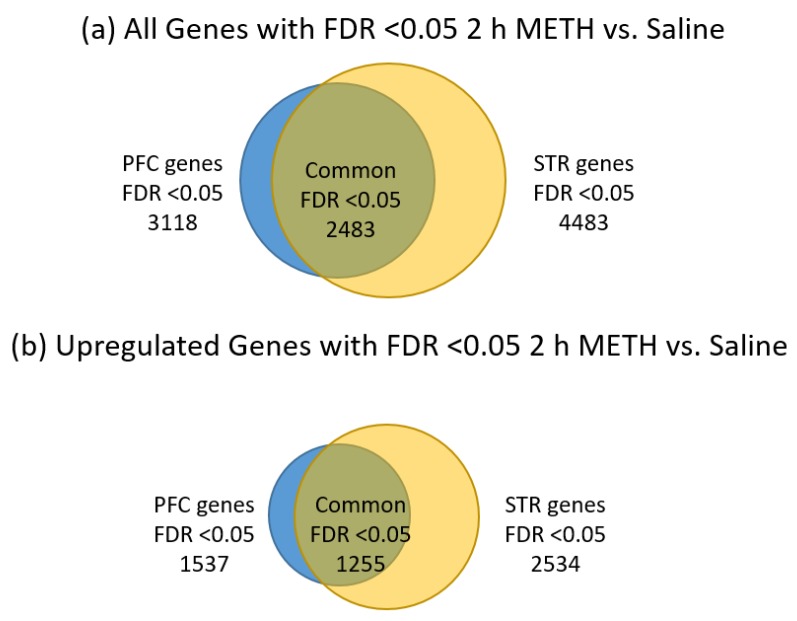
Distribution of significantly-changed genes FDR <0.05, 2 h post METH, in the STR and PFC. (**a**) all significantly changed genes; (**b**) significantly upregulated genes; (**c**) significantly downregulated genes. The relative number of significantly-changed genes are represented by the size of the circles. Genes significantly-changed in the prefrontal cortex are shown by blue circles while those from the striatum are shown by yellow circles. Genes shared by both regions are depicted by the overlapping area between the two circles in an olive color. Note that the striatum has the larger number of gene changes compared to the prefrontal cortex and that about 80% of the genes changed by METH in the PFC were also changed in the STR.

**Figure 3 brainsci-09-00340-f003:**
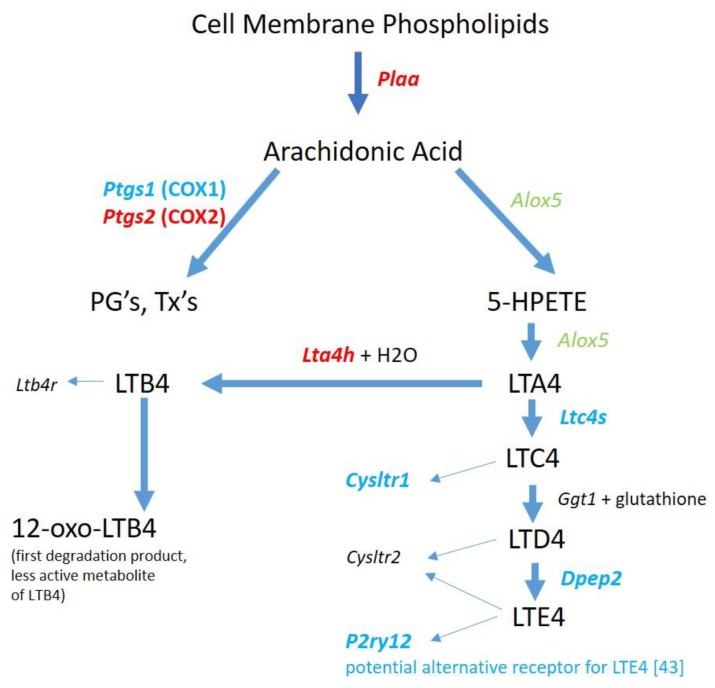
Genes for prostaglandin (PG) and leukotrienes (LT) biosynthetic enzymes and LT receptors in 2 h post METH STR microglia. METH treatment appears to direct the release of arachidonic acid from cell membrane phospholipids toward synthesis of prostaglandins and leukotriene B4 which are known to have roles in inflammation. Downregulation or lack of detection of RNA for all leukotriene receptors [43] suggests that microglia are less responsive to leukotrienes post-binge METH. Changes in gene expression for enzymes and receptors are denoted by italic text and color: Black = not detected. Green = no significant change. Red = significant upregulation. Blue = significant downregulation. Gene expression data can be found in Appendix A.

**Figure 4 brainsci-09-00340-f004:**
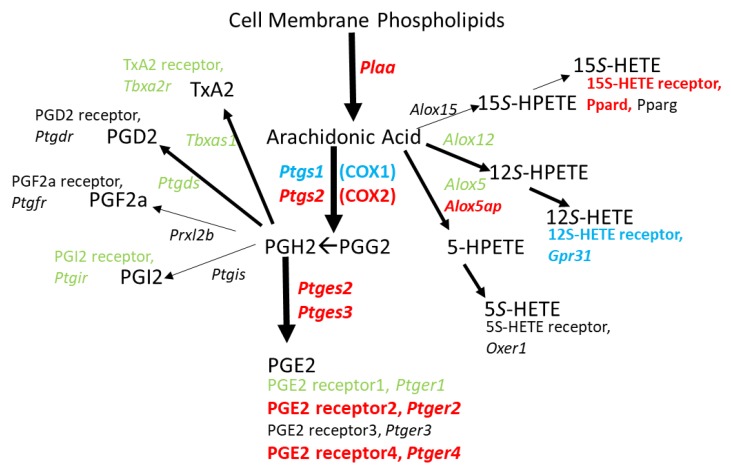
Genes for biosynthetic enzymes and receptors for PG’s, TxA2, and HETE’s in 2 h post METH STR microglia. METH treatment seems to direct the release of arachidonic acid from cell membrane phospholipids particularly toward synthesis of PGE2 over any of the other possible cyclooxygenase- or lipoxygenase-derived products. In addition, METH provides a means by which PGE2 can act in an autocrine fashion with the concurrent upregulation of genes for PGE2 receptors EP2 and EP4. Changes in gene expression are denoted by italic text and color: Black = not detected. Green = no significant change. Red = significant upregulation. Blue = significant downregulation. Gene expression data can be found in Appendix A.

**Figure 5 brainsci-09-00340-f005:**
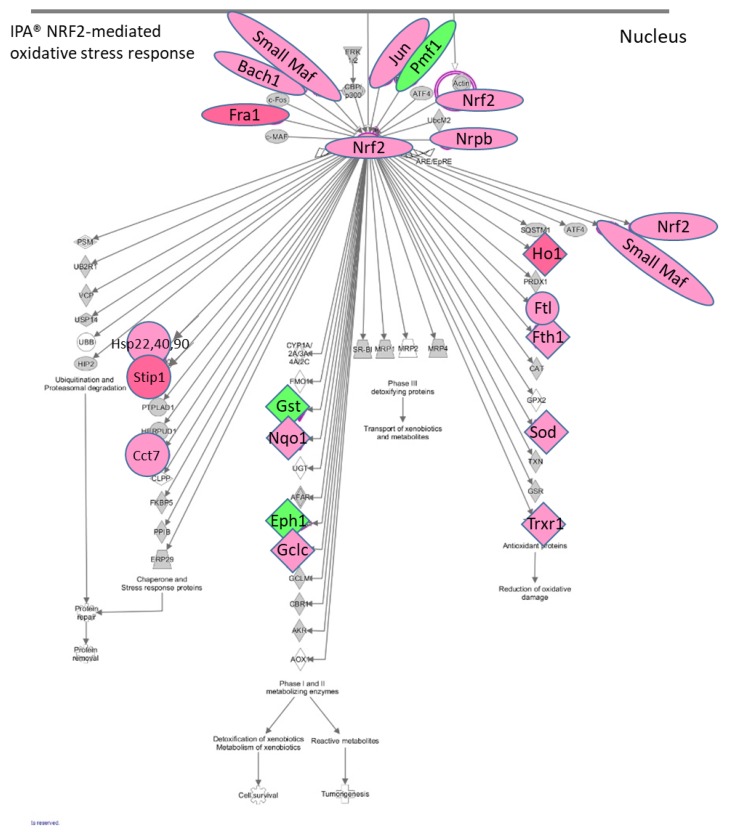
RNAseq genes associated with NRF2-mediated oxidative stress response via IPA^®^. Toxic doses of METH can produce oxidative stress in the rodent brain. In the rat striatum 2 h post METH, IPA^®^ showed that the upregulation of a number of genes associated with oxidative stress are under the control of NRF2, which is itself upregulated by METH. Among the upstream mediators which can initiate this pathway are drugs, cytokines prostaglandins, and bacterial infections. RNAseq data showed that *Il1a*, *Il1b*, and the enzymes which can enhance synthesis of PGE2 are also upregulated by METH and could contribute to initiating the NRF2-mediated oxidative stress response. Red symbols = upregulated genes. Green symbols = downregulated genes. Gene expression data can be found in Appendix A.

**Figure 6 brainsci-09-00340-f006:**
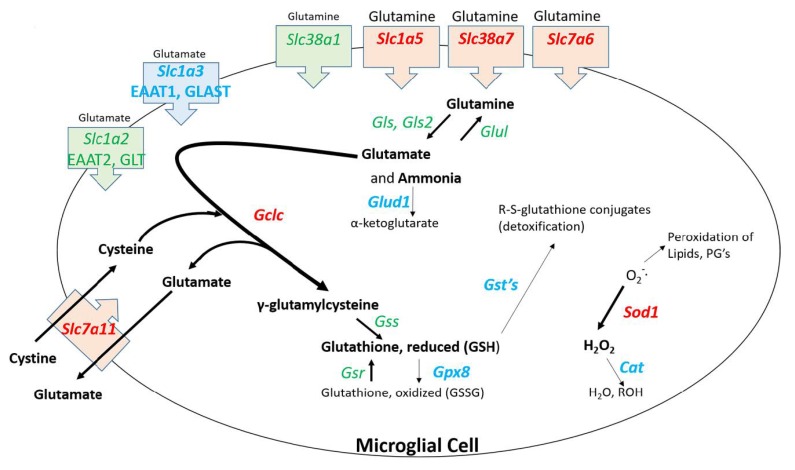
Hypothesized pathway and consequences of glutamine uptake in 2 h post METH STR microglia. Binge dose METH leads to neuronal release of glutamate. Glutamate is taken up by astrocytes, converted to glutamine, and released to the extracellular space. Upregulation of genes for several glutamine transporters suggests the enhanced uptake of glutamine by microglia. The potential increased levels of glutamine would stoichiometrically be converted to glutamate and ammonia [83]. The resulting enhanced levels of glutamate may be met by the enhanced expression of *Gclc*, the gene for the first rate-limiting enzyme in glutathione synthesis, and enhanced expression of *Slc7a11*, the gene for a cystine/glutamate transporter. This combination may drive the enhanced synthesis of γ-glutamylcysteine and the reduced form of glutathione (GSH). The utility of GSH to participate in detoxifying reactive species may be thwarted by the concurrent downregulation of genes for several glutathione-S-transferases (GSTs). Enhanced expression of *Sod1* via the IPA-predicted enhancement NRF2 signaling may serve to partially reduce the amounts of METH-generated reactive oxygen species. However, there is incomplete conversion of the resulting toxic hydrogen peroxide to water and alcohol and accompanying generation of oxidized glutathione due to the downregulation of genes for catalase (*Cat*) and glutathione peroxidase 8 (*Gpx8*), respectively. Other sources of toxic species resulting from enhanced uptake of glutamine by microglia could be the enhanced presence/release of ammonia after the conversion of glutamine to glutamate via the concurrent downregulation of glutamate dehydrogenase 1 (*Glud1*) as well as the enhanced export of glutamate via upregulated *Slc7a11*. Changes in gene expression are denoted by text color and font for the gene symbol: Green = no significant change. Red, bold = significant upregulation. Blue, bold = significant downregulation. Gene expression data can be found in Appendix A.

**Table 1 brainsci-09-00340-t001:** Number of genes with differential expression methamphetamine (METH) vs. saline, false discovery rate (FDR) < 0.05 unpaired t-test. PFC: prefrontal cortex; STR: striatum; FC: fold change.

	2 h Post METH				3 Day Post METH			
**Totals**	**PFC Genes: 10,496**		**STR Genes: 10,839**		**PFC Genes: 9991**		**STR Genes: 10,612**	
Considering All FC’s with FDR < 0.05.								
**Totals**	**PFC Genes: 3118**		**STR Genes: 4483**		**PFC Genes: 0**		**STR Genes: 714**	
	Up	Down	Up	Down	Up	Down	Up	Down
	1537	1581	2435	2048	0	0	400	314
Considering FC’s > 2 or <−2 with FDR < 0.05.								
**Totals**	**PFC Genes: 934**		**STR Genes: 1531**		**PFC Genes: 0**		**STR Genes: 78**	
	Up	Down	Up	Down	Up	Down	Up	Down
	436	498	840	691	0	0	41	37

**Table 2 brainsci-09-00340-t002:** Genes common to PFC and STR with opposite differential expression METH vs. saline, 2 h time point, FDR <0.05 unpaired t-test.

Symbol	Gene Name	Fold Change PFC 2 h	Fold Change STR 2 h
*Grik5*	Glutamate Ionotropic Receptor Kainate Type Subunit 5	3.39	−2.74
*Plau*	Plasminogen Activator, Urokinase	−2.25	4.04
*Itgb3*	Integrin Subunit Beta 3	−1.58	1.52
*Tmem120a*	Transmembrane Protein 120A	−1.30	1.40
*Msto1*	Misato Mitochondrial Distribution and Morphology Regulator 1	−1.31	1.29

**Table 3 brainsci-09-00340-t003:** Genes in STR with significant differential expression at both 2 h and 3 day time points, fold changes <−2 or >+2.

Genes Upregulated at Both Time Points			
Symbol	Gene Name	Fold Change 2 h	Fold Change 3 day
*Slc6a12*	solute carrier family 6 member 12	16.60	2.75
*Csf1*	colony stimulating factor 1	12.52	2.41
*Cd5*	Cd5 molecule	3.40	2.29
*Fn1*	fibronectin 1	3.14	2.65
*Adgrg5*	adhesion G protein-coupled receptor G5	2.81	2.30
*LOC685157*	similar to paired immunoglobin-like type 2 receptor beta	2.67	3.15
**Genes Downregulated at Both Time Points**			
Symbol	Gene Name	Fold Change 2 h	Fold Change 3 day
*Gpr12*	G protein-coupled receptor 12	−5.74	−2.72
*Gba3*	glucosidase, beta, acid 3	−5.40	−4.78
*Ripply3*	ripply transcriptional repressor 3	−2.88	−2.13
*Sntb1*	syntrophin, beta 1	−2.87	−2.00
*Tmem53*	transmembrane protein 53	−2.49	−2.18
*Dscc1*	DNA replication and sister chromatid cohesion 1	−2.44	−2.36
**Genes Upregulated at 2 h, Downregulated at 3 Days**			
Symbol	Gene Name	Fold Change 2 h	Fold Change 3 day
*Hbegf*	heparin-binding EGF-like growth factor	17.55	−3.16
*Bcl2*	BCL2, apoptosis regulator	3.74	−2.14
*Avpr1a*	arginine vasopressin receptor 1A	2.71	−2.15
*Spata22*	spermatogenesis associated 22	2.40	−4.78
*Ccnd2*	cyclin D2	2.31	−2.75
**Genes Downregulated at 2 h, Upregulated at 3 Days**			
Symbol	Gene Name	Fold Change 2 h	Fold Change 3 day
*Islr*	immunoglobulin superfamily containing leucine-rich repeat	−9.94	3.02
*LOC689081*	similar to cystatin E2	−6.14	2.23
*Fndc1*	fibronectin type III domain containing 1	−3.33	3.80
*Hfe*	hemochromatosis	−2.62	2.21

**Table 4 brainsci-09-00340-t004:** Ingenuity^®^ Pathway Analysis of significantly changed genes within each brain region and time point.

	Top Canonical Pathways	Top Upstream Regulators	Top Molecular and Cellular Functions
STR 2 h post-METH, FDR <0.05	NRF2-mediated oxidative stress response	Lipopolysaccharide	Cell death and survival
	Aryl hydrocarbon receptor signaling	PDGF BB	Cellular movement
	Unfolded protein response	Beta-estradiol	Cellular development
	Aldosterone signaling in epithelial cells	HSF1	Cellular growth and proliferation
	LXR/RXR activation	TNF	Cell morphology
PFC 2 h post-METH, FDR <0.05	Aldosterone signaling in epithelial cells	Lipopolysaccharide	Cellular movement
	Unfolded protein response	IFNG	Cell death and survival
	Atherosclerosis signaling	TGFB1	Protein synthesis
	Granulocyte adhesion and diapedesis	IL1B	Cellular function and maintenance
	IL-10 signaling	PDGF BB	Cellular development
STR 3 day post-METH, FDR <0.05	EIF2 signaling	TCR	RNA post-transcriptional modification
	Regulation of eIF4 and p70S6K signaling	ERBB2	Protein synthesis
	mTOR signaling	DOT1L	Cell death and survival
	Type I diabetes mellitus signaling	EBI3	Gene expression
	T helper cell differentiation	DRAP1	DNA replication, recombination, and repair
PFC 3 day post-METH, FDR <0.05	N/A	N/A	N/A

**Table 5 brainsci-09-00340-t005:** Ingenuity^®^ Pathway Analysis of significantly changed genes in STR at both 2 h and 3 days post-METH.

	Top Canonical Pathways	Top Upstream Regulators	Top Molecular and Cellular Functions
Upregulated at both 2 h and 3 day post-METH	EIF2 signaling	TCR	RNA post-transcriptional modification
	Regulation of eIF4 and p70S6K signaling	DOT1L	Protein synthesis
	mTOR signaling	HSP90B1	Gene expression
	Hepatic fibrosis/hepatic stellate cell activation	miR-542-3p (miRNAs w/seed GUGACAG)	Cell morphology
	Complement system	ELAVL1	Cell-to-cell signaling and interaction
Downregulated at both 2 h and 3 day post-METH	Tryptophan degradation to 2-amino-3-carboxymuconate semialdehyde	BHLHA15	Cellular assembly and organization
	NAD biosynthesis II (from tryptophan)	SOX11	DNA replication, recombination, and repair
	Leukotriene biosynthesis	ERBB2	Cell cycle
	-Glutamyl cycle	NAMPT	Cellular growth and proliferation
	Mismatch repair in eukaryotes	CERS6	Cell morphology
Upregulated at 2 h and downregulated at 3 day	-Linolenate biosynthesis II (animals)	FAM103A1	Cellular assembly and organization
	Myc mediated apoptosis signaling	IKBKE	DNA replication, Recombination and repair
	Glycine biosynthesis I	PIM1	Cell morphology
	Pancreatic adenocarcinoma signaling	RELB	Cell death and survival
	S-adenosyl-L-methionine biosynthesis	NFKB2	Cell cycle
Downregulated at 2 h and upregulated at 3 day	NAD salvage pathway II	NDRG1	Lipid metabolism
	IL-22 signaling	COL18A1	Small molecule biochemistry
	Natural killer cell signaling	CDK4	Cell morphology
	PI3K/AKT signaling	PDLIM2	Cellular function and maintenance
	Interferon signaling	CLDN7	Cell death and survival

**Table 6 brainsci-09-00340-t006:** RNAseq data for top 25 genes enriched in microglia [32]. (U) = unique to microglia vs. oligodentrocyes, astrocytes, and neurons. Bold text = change in expression FDR <0.05 METH vs. Saline treatment. ND = not detected. NS = not significant.

Symbol	Gene Name		STR Average log2cpm Saline (*n* = 6), METH (*n* = 8)	STR Fold Change, FDR 2 h METH vs. Saline	PFC Average log2cpm Saline (*n* = 4), METH (*n* = 5)		PFC Fold Change, FDR 2 h METH vs. Saline
***P2ry12* (U)**	**purinergic receptor P2Y12**		**10.88, 9.77**	**−2.01, 5.0 × 10^−5^**	**10.93, 10.23**		**−1.57, 4.5 × 10^−5^**
*Tmem119* (U)	transmembrane protein 119		11.24, 11.04	−1.19, 0.39 (NS)	11.28, 11.38		1.11, 0.72 (NS)
*Fcrls* (Fcrl2) (U)	Fc receptor-like 2		10.04, 9.81	−1.08, 0.83 (NS)	10.51, 10.44		−1.11, 0.81 (NS)
***Olfml3* (U)**	**olfactomedin-like 3**		**1.18, −0.29**	**−2.78, 3.7 × 10^−5^**	**0.98, −0.08**		**−2.15, 0.04**
*Hexb* (U)	hexosaminidase subunit beta		12.68, 12.48	−1.14, 0.23 (NS)	12.77, 12.49		−1.20, 0.11 (NS)
***Ctss***	**cathepsin S**		**14.68, 15.04**	**1.30, 0.01**	14.70, 15.05		1.29, 0.06 (NS)
***C1qb***	**complement C1q B chain**		**12.73, 13.23**	**1.44, 0.001**	**12.66, 13.36**		**1.66, 2.2 × 10^−4^**
*C1qa*	**complement C1q A chain**		**12.57, 12.82**	1.21, 0.11 (NS)	**12.53, 12.89**		**1.31, 0.04**
*Csf1r*	colony stimulating factor 1 receptor		13.48, 13.62	1.10, 0.54 (NS)	13.42, 13.51		1.03, 0.85 (NS)
***P2ry13***	**purinergic receptor P2Y13**		**9.27, 8.11**	**−2.04, 1.1 × 10^−4^**	**9.16, 8.53**		**−1.46, 0.01**
***Cx3cr1***	**C-X3-C motif chemokine receptor 1**		**12.27, 11.55**	**−1.50, 0.01**	12.17, 11.73		−1.31, 0.11 (NS)
***Gpr34***	**G protein-coupled receptor 34**		**11.39, 10.51**	**−1.72, 0.02**	**11.43, 10.54**		**−1.76, 0.004**
*C1qc*	complement C1q C chain		11.01, 11.30	1.20, 0.15 (NS)	10.85, 11.40		1.49, 0.11 (NS)
***Mafb***	**MAF bZIP transcription factor B**		**6.55, 7.31**	**2.06, 0.008**	6.41, 6.74		1.27, 0.24 (NS)
*Tgfbr1* (U)	transforming growth factor, beta receptor 1		10.46, 10.39	−1.04, 0.80 (NS)	10.60, 10.50		−1.08, 0.62 (NS)
*Fcgr1*	Fc fragment of IgG receptor Ia		7.45, 7.15	−1.17, 0.33 (NS)	7.40, 7.23		−1.12, 0.50 (NS)
***Entpd1***	**ectonucleoside triphosphate diphosphohydrolase 1**		**6.39, 5.52**	**−1.74, 3.0 × 10^−4^**	**6.37, 5.86**		**−1.42, 0.02**
***Ltc4s***	**leukotriene C4 synthase**		**8.11, 6.91**	**−2.14, 1.4 × 10^−5^**	**8.11, 7.35**		**−1.61, 8.5 × 10^−4^**
***Csf3r***	**colony stimulating factor 3 receptor**		**8.30, 9.58**	**2.50, 1.2 × 10^−7^**	**8.24, 9.53**		**2.42, 8.6 × 10^−9^**
***Il10ra***	**interleukin 10 receptor subunit alpha**		**8.35, 7.75**	**−1.52, 2.0 × 10^−4^**	8.43, 8.07		−1.23, 0.06 (NS)
*Egr1*	early growth response 1		8.36, 8.81	1.91, 0.23 (NS)	8.30, 7.67		−1.39, 0,45 (NS)
*Siglech*	sialic acid binding Ig like lectin H		ND	ND	ND		ND
*Ccl2*	C-C motif chemokine ligand 2		5.57, 6.34	2.15, 0.17 (NS)	5.66, 5.62		1.03, 0.97 (NS)
*Lag3*	lymphocyte activating 3		3.84, 3.80	1.01, 0.98 (NS)	3.83, 3.65		−1.16, 0.63 (NS)
*Fos*	FBJ osteosarcoma oncogene		7.21, 7.31	1.09, 0.92 (NS)	8.45, 8.37		1.14, 0.86 (NS)
**Average log2cpm**							
**STR, Saline 10,839 Genes**		**STR, METH 10,839 Genes**		**PFC, Saline 10,496 Genes**		**PFC, METH 10,496 Genes**	
3.93		3.91		4.13		4.14	
STR, Saline Top 25 genes		STR, METH Top 25 genes		PFC, METH Top 25 genes		PFC, METH Top 25 genes	
9.34		9.16		9.38		9.25	

**Table 7 brainsci-09-00340-t007:** Average log2cpm of genes unique to mouse brain microglia or peritoneal macrophages as per Hickman et al. [34].

	STR Average Saline Microglial log2cpm (*n* = 6)	STR Average METH Microglial log2cpm (*n* = 8)	PFC Average Saline Microglial log2cpm (*n* = 4)	PFC Average METH Microglial log2cpm (*n* = 5)
Mouse Microglial Unique Genes	8.06	7.53	8.49	7.94
Mouse Macrophage Unique Genes	1.51	1.95	1.38	2.32

**Table 8 brainsci-09-00340-t008:** Literature-cited M1 or M2 microglial genes/proteins compared to RNAseq data. Green-font rows indicate significant change in expression that is directionally in agreement with literature citations. Yellow-font rows indicate significantly changed RNAseq genes that are closely related to literature-cited genes. ND = not detected. NS = not significant.

Symbol	Gene Name	M1/M2	Source	Brain Region	Average Log2 cpm Sal, METH	Fold Change	FDR
*Ccl2 (MCP-1)*	C-C motif chemokine ligand 2	M1	Zhou	STR	5.57, 6.34	2.15	0.17 (NS)
PFC	5.66, 5.62	1.03	0.97 (NS)
*Ccl5 (RANTES)*	C-C motif chemokine ligand 5	M1	Chhor	STR	ND, ND	ND	
PFC	ND, ND	ND	
*Ccr7*	C-C Motif Chemokine Receptor 7	M1	Barakat	STR	ND, ND	ND	
PFC	−4.59, −2.62	1.84	0.74 (NS)
*Fcgr3a (Cd16)*	Fc fragment of IgG, low affinity IIIa, receptor	M1	Barakat, Zhou	STR	8.12, 7.25	−1.78	8.5 × 10^−8^
PFC	8.01, 7.01	−1.97	1.5 × 10^−9^
*Fcer2 (Cd23)*	Fc Fragment Of IgE Receptor II	M1	Barakat	STR	ND, ND	ND	
PFC	ND, ND	ND	
*Fcgr2a (Cd32)*	Fc fragment of IgG, low affinity IIa, receptor	M1	Barakat, Zhou, Peng	STR	7.16, 7.79	1.58	9.6 × 10^−7^
PFC	7.25, 7.96	1.67	1.7 × 10^−5^
*Cd40 (Tnfr)*	Cd40 molecule	M1	Barakat	STR	6.08, 5.45	−1.46	0.13 (NS)
PFC	6.29, 5.30	−1.97	1.9 × 10^−9^
*Cd80*	Cd80 molecule	M1	Barakat	STR	3.24, 3.62	1.34	0.049
PFC	3.30, 3.68	1.32	0.14 (NS)
*Cd86*	Cd86 molecule	M1	Barakat, Zhou, Peng	STR	6.68, 6.47	−1.13	0.29 (NS)
PFC	6.54, 6.57	1.11	0.41 (NS)
*Cxcl1*	C-X-C motif chemokine ligand 1	M1	Chhor	STR	−3.53, −0.28	6.3	0.01
PFC	−2.30, 0.40	5.09	0.001
*Adgre1 (F4/80)*	adhesion G protein-coupled receptor E1	M1	Barakat	STR	10.32, 10.05	−1.2	0.06 (NS)
PFC	10.47, 10.35	−1.1	0.47 (NS)
*Il1a*	interleukin 1 alpha	M1	Chhor	STR	6.84, 7.93	3.98	0.001
PFC	6.76, 7.50	2.16	0.03
*Il1b*	interleukin 1 beta	M1	Barakat, Chhor	STR	3.83, 5.97	4.82	1.2 × 10^−5^
PFC	3.85, 6.28	5.57	7.6 × 10^−13^
*Il2*	Interleukin 2	M1	Chhor	STR	ND, ND	ND	
PFC	ND, ND	ND	
*Il6*	interleukin 6	M1	Barakat, Chhor, Zhou	STR	3.05, 2.62	−1.26	0.35 (NS)
PFC	3.13, 3.33	1.17	0.58 (NS)
*Il12*	interleukin 12	M1	Barakat, Chhor	STR	5.04, 4.52	−1.36	0.045
PFC	4.89, 4.47	−1.25	0.32 (NS)
*Ifng*	interferon gamma	M1	Chhor	STR	ND, ND	ND	
PFC	ND, ND	ND	
*Nos2 (iNos)*	nitric oxide synthase 2	M1	Barakat, Chhor	STR	ND, ND	ND	
PFC	ND, ND	ND	
*Hla-d* (Mhc II)*	major histocompatibility class II	M1	Barakat, Peng	STR	ND, ND	ND	
PFC	ND, ND	ND	
*Tnf*	tumor necrosis factor	M1	Barakat, Chhor, Zhou	STR	1.19, 0.89	−1.26	0.49 (NS)
PFC	0.96, 0.46	−1.26	0.65 (NS)
Gene related to *Tnf* from RNAseq Data							
*Tnfsf9*	tumor necrosis factor superfamily member 9			STR	0.33, 4.29	19.53	5.7 × 10^−11^
PFC	0.23, 3.63	11.45	2.5 × 10^−8^
*Ptgs2* (COX2)	prostaglandin-endoperoxide synthase 2	M1/M2	Chhor	STR	−4.73, −0.62	47.3	0.002
PFC	ND, ND	ND	
*Arg1*	Arginase 1	M2	Cherry, Chhor	STR	ND, ND	ND	
PFC	ND, ND	ND	
*Ccl2 (MCP-1)*	C-C motif chemokine ligand 2	M2	Chhor	STR	5.57, 6.34	2.15	0.17 (NS)
PFC	5.66, 5.62	1.03	0.97 (NS)
*Ccr2 (MCP-1-R)*	C-C Motif Chemokine Receptor 2	M2	Chhor	STR	ND, ND	ND	
PFC	ND, ND	ND	
*Cd163*	CD163 molecule	M2	Barakat, Cherry	STR	ND, ND	ND	
PFC	−2.16, 2.13	7.21	0.11 (NS)
*Msr1 (Cd204) *	macrophage scavenger receptor 1	M2	Barakat	STR	1.38, 2.58	2.55	3.4 × 10^−4^
PFC	1.06, 3.01	4.18	2.6 × 10^−6^
*Cd209*	CD209 molecule	M2	Barakat	STR	ND, ND	ND	
PFC	ND, ND	ND	
*Cd36*	CD36 molecule	M2	Barakat	STR	−2.79, −2.13	1.12	0.88 (NS)
PFC	−2.15, −0.79	1.91	0.42 (NS)
*Cd68*	Cd68 molecule	M2	Barakat	STR	6.83, 6.68	−1.11	0.38 (NS)
PFC	6.70, 6.49	−1.14	0.28 (NS)
*Cxcr3*	C-X-C motif chemokine receptor 3	M2	Chhor	STR	2.49, 1.61	−1.53	0.21 (NS)
PFC	2.79, 2.15	−1.54	0.01
* Clec7a (Dectin 1)*	C-type lectin domain family 7, member A	M2	Barakat, Cherry	STR	2.34, 2.59	1.38	0.35 (NS)
PFC	1.22, 2.75	3.02	0.006
*Retnlb (Fizz1)*	Resistin Like Beta	M2	Barakat, Cherry, Chhor	STR	ND, ND	ND	
PFC	ND, ND	ND	
*Lgals3 (Gal-3)*	galectin 3	M2	Chhor	STR	0.79, 3.06	6.2	3.0 × 10^−4^
PFC	0.13, 2.49	6.7	6.8 × 10^−4^
*Csf3 (G-CSF)*	Colony Stimulating Factor 3	M2	Chhor	STR	ND, ND	ND	
PFC	ND, ND	ND	
*Csf2 (GM-CSF)*	Colony Stimulating Factor 2	M2	Chhor	STR	ND, ND	ND	
PFC	ND, ND	ND	
Gene related to *Csf3* from RNAseq Data							
*Csf1 (M-CSF) *	Colony Stimulating Factor 1			STR	1.54, 4.91	12.52	7.6 × 10^−9^
PFC	1.71, 4.14	7.1	7.3 × 10^−7^
*Igf1*	insulin-like growth factor 1	M2	Chhor	STR	4.81, 5.26	1.41	0.01
PFC	4.87, 5.57	1.65	0.004
*Il1rn (Il1ra) *	interleukin 1 receptor antagonist	M2	Chhor	STR	−0.40, 1.39	4.04	0.002
PFC	−0.29, 1.19	3.43	0.4 (NS)
Genes related to *Il1rn* from RNAseq Data							
*Il1rl1* (receptor for IL33 with Il1RAP)	interleukin 1 receptor-like 1			STR	6.31, 5.82	−1.39	0.01
PFC	6.26, 5.87	−1.3	0.08 (NS)
*Il1rap*	interleukin 1 receptor accessory protein			STR	3.49, 3.94	1.36	0.004
PFC	3.29, 3.55	1.19	0.29 (NS)
*Il1r2* (IL1b decoy receptor)	interleukin 1 receptor type 2			STR	−1.75, 3.59	39.1	1.9 × 10^−10^
PFC	−3.37, 4.48	131.46	2.0 × 10^−10^
*Il4*	Interleukin 4	M2	Barakat, Chhor	STR	ND, ND	ND	
PFC	ND, ND	ND	
*Il4r*	Interleukin 4 Receptor	M2	Chhor	STR	8.12, 8.41	1.25	0.21 (NS)
PFC	8.11, 8.39	1.23	0.16 (NS)
*Il10*	Interleukin10	M2	Barakat, Chhor	STR	ND, ND	ND	
PFC	ND, ND	ND	
Genes related to *Il10* from RNAseq Data							
*Il10ra*	interleukin 10 receptor subunit alpha			STR	8.35, 7.75	−1.52	2.0 × 10^−4^
PFC	8.34, 8.07	−1.23	0.06 (NS)
*Il10rb*	interleukin 10 receptor subunit beta			STR	7.98, 7.53	−1.34	6.0 × 10^−4^
PFC	7.94, 7.62	−1.22	0.04
*Il13*	Interleukin13	M2	Barakat	STR	ND, ND	ND	
PFC	ND, ND	ND	
Gene related to *Il13* from RNAseq Data							
*Il13ra1*	interleukin 13 receptor subunit alpha 1			STR	3.77, 4.01	1.18	0.19 (NS)
PFC	3.77, 4.48	1.6	1.7 × 10^−3^
*Clec10a (Cd301)*	C-type lectin domain family 10, member A	M2	Cherry	STR	−0.89, −0.36	−1.19	0.75 (NS)
PFC	1.78, 3.67	4.08	0.08 (NS)
*Mrc1 (Cd206)*	mannose receptor, C type 1	M2	Barakat Cherry, Chorr, Zhou, Peng	STR	−0.86, 0.66	2.72	0.009
PFC	0.83, 3.77	6.39	0.04
*Socs3*	suppressor of cytokine signaling 3	M2	Chhor	STR	5.31, 5.49	1.04	0.91 (NS)
PFC	5.71, 6.13	1.34	0.29 (NS)
*SphK1*	Sphingosine Kinase 1	M2	Chhor	STR	ND, ND	ND	
PFC	ND, ND	ND	
*SphK2*	sphingosine kinase 2	M2	Chhor	STR	5.90, 6.02	1.1	0.37 (NS)
PFC	6.05, 6.23	1.14	0.30 (NS)
*TGFb*, subtypes below		M2	Barakat, Chhor				
*TGFb1*	transforming growth factor, beta 1			STR	6.14, 6.14	−1.09	0.62 (NS)
PFC	6.34, 6.40	1	1.00 (NS)
*TGFb2*	transforming growth factor, beta 2			STR	−1.15, −1.34	−1.14	0.80 (NS)
PFC	ND, ND	ND	
*TGFb3*	transforming growth factor, beta 3			STR	−0.63, −1.79	−1.27	0.72 (NS)
PFC	−0.93, −1.46	−1.24	0.77 (NS)
*Trem2*	triggering receptor expressed on myeloid cells 2	M2	Cherry	STR	10.50, 9.06	−2.58	1.30 × 10^−8^
PFC	10.59, 8.88	−3.18	6.1 × 10^−14^
*Chi3l3 (Ym1)*	chitinase 3 like 3	M2	Barakat, Cherry, Chhor	STR	ND, ND	ND	
PFC	ND, ND	ND

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
