# Peer review of "Evaluation of Microglia/Macrophage Cells from Rat Striatum and Prefrontal Cortex Reveals Differential Expression of Inflammatory-Related mRNA after Methamphetamine"

_brainsci, 2019, doi:10.3390/brainsci9120340_

Round 1
Reviewer 1 Report
I have read this article focused on RNA changes in the brain produced by a prior history of binge-like Meth exposure in rats. Using RNAseq approaches, the authors provide novel evidence that CD11b/c+ cells (to index resident microglia and infiltrating marcrophages) from the STR and PFC exhibit changes in mRNA expression indicative of a pro-inflammatory state, with some evidence of an opponent anti-inflammatory process. The mRNA changes were more robust in STR than PFC, and more numerous in early (2-h) versus later (3 days) withdrawal. Overall, this report is extremely well-written; the Introduction is succinct, the Methods are clearly described, the Results (while numerous) are presented in a clear and comprehensible manner and the Discussion, although quite lengthy, is very thorough and thoughtful. In all honesty, this report was a pleasure to read and a genuine learning experience. I have very few comments or criticisms.
While the authors define M1 and M2 phenotypes very well in the Introduction, it is not clear what M2 means in the abstract. Please add in a bit of detail as was done to help the reader understand the meaning of M1 in the abstract. I was curious why the authors chose to perfuse the rats with PBS prior to brain removal? In my experience, fresh/non-perfused tissue is typically employed when conducting mRNA measurements as the administration of anesthetics and perfusion procedures can induce IEG changes and phosphorylation events that might impact mRNA expression. The font size of the text in Figure 5 is quite small. I recommend trying to enlarge the font. Likewise, the font size of the text in Figure 6 could also be enlarged, particularly for the red and blue genes, which are the genes exhibiting changes in the present study.Author Response
Reviewer 1. We appreciate the comments from the reviewer.
While the authors define M1 and M2 phenotypes very well in the Introduction, it is not clear what M2 means in the abstract. Please add in a bit of detail as was done to help the reader understand the meaning of M1 in the abstract.
We have reworded the abstract to provide a definition of M2 in contrast to M1 as a pro-inflammatory state.
I was curious why the authors chose to perfuse the rats with PBS prior to brain removal? In my experience, fresh/non-perfused tissue is typically employed when conducting mRNA measurements as the administration of anesthetics and perfusion procedures can induce IEG changes and phosphorylation events that might impact mRNA expression.
We chose to perfuse with saline prior to brain extraction to reduce the contribution of circulating macrophages contained in the brain’s capillaries as vs. those which had potentially exited the circulation and infiltrated the brain tissue. All rats (Meth and saline treated rats) were anesthetized and underwent saline perfusion to control for the possible effects of anesthetics or saline perfusion across drug groups. Thus, any differences can be attributed to the effects of Meth.
The font size of the text in Figure 5 is quite small. I recommend trying to enlarge the font. Likewise, the font size of the text in Figure 6 could also be enlarged, particularly for the red and blue genes, which are the genes exhibiting changes in the present study.
We acknowledge the font for the unchanged genes in Figure 5 is very small. This is the font provided by the IPA output. Since these genes were not changed by the treatment, we did not try to change the legibility of those gene symbols. We overlaid larger shapes and fonts for only the genes that were changed to draw attention to those genes in Figure 5 and focus on those genes for the Discussion.
We have amended Figure 6 with larger font for all the red, blue and green coded genes. The changed genes in red and blue are in bold font while the genes in green are in standard font.
Reviewer 2 Report
The paper of Kays and Yamamoto shows in detail the effect of methamphetamine on microglia in rat brain regions. The data are interesting and the manuscript can be accepted with some modifications
1- specify the number of rats per group.
For how many rats the RNA sequencing experiments were conducted.
Are the data representative or an average of the results obtained in different animals?
PLease specify in the methods
2- The discussion is too complex and long to read (it almost seems like a mini review).
Please shorten it only by picking up the thread of the important results obtained and where possible to mention recent review rather than individual dated, manuscript.
Author Response
We appreciated the comments of the reviewer.
1- specify the number of rats per group.
We acknowledge there is some inconsistency in the listing of group size in the original manuscript. The number of rats per group are listed in the column headings of Tables 6 and 7. The number of rats per group is missing from the column headings in Table 8 since average values for STR and PFC share the same column but separate rows. The term “Ave” is used as an abbreviation for average in the column headings of Tables 6, 7, and 8, but the abbreviation is not defined.
For how many rats the RNA sequencing experiments were conducted.
We have amended the Methods section to include the number of rats per group.
Are the data representative or an average of the results obtained in different animals? PLease specify in the methods
We have spelled out the word average in the column headings.
If spelling out “average” in the column heading is not suitable for the spacing of the text in the column, please let us know and we can instead use “Ave” in the column headings and add the text below to the end of the table legends for Tables 6, 7, and 8: Ave = average.
The discussion is too complex and long to read (it almost seems like a mini review).
Please shorten it only by picking up the thread of the important results obtained and where possible to mention recent review rather than individual dated, manuscript.
We have significantly shortened the Discussion to focus on METH’s effect on genes associated with arachidonic acid metabolism and glutamine conversion.